# Genetic Susceptibility and Genetic Variant-Diet Interactions in Diabetic Retinopathy: A Cross-Sectional Case–Control Study

**DOI:** 10.3390/nu17182983

**Published:** 2025-09-17

**Authors:** Sunmin Park, Suna Kang, Donghyun Jee

**Affiliations:** 1Department of Food and Nutrition, Obesity/Diabetes Research Center, Hoseo University, Asan 31499, Republic of Korea; roypower003@naver.com; 2Division of Vitreous and Retina, Department of Ophthalmology, St. Vincent’s Hospital, College of Medicine, The Catholic University of Korea, Suwon 16247, Republic of Korea

**Keywords:** diabetic retinopathy, polygenic risk scores, neuronal signaling, eating duration, alcohol

## Abstract

**Background/Objectives:** Diabetic retinopathy is a leading cause of blindness in diabetic patients, with disease susceptibility influenced by both genetic and environmental factors. This study aimed to identify novel genetic variants associated with DR and evaluate interactions between polygenic risk scores (PRS) and lifestyle factors in a Korean diabetic cohort. **Methods:** After excluding subjects with non-diabetic retinopathy eye diseases (n = 2519), we analyzed data from 50,361 non-diabetic controls, 4873 diabetic participants without retinopathy (DM-NR), and 165 with diabetic retinopathy (DM-DR). We conducted genome-wide association studies comparing DM-NR and DM-DR groups, performed generalized multifactor dimensionality reduction (GMDR) analysis for epistatic interactions, developed unweighted PRS models, and examined PRS–lifestyle interactions using two-way analysis of covariance. **Results:** DM-DR prevalence showed strong associations with metabolic syndrome and its components. Five novel genetic variants were identified: *ABCA4*_rs17110929, *MMP2-AS1*_rs2576531, *FOXP1*_rs557869288, *MRPS33*_rs1533933, and *DRD2*_rs4936270. A significant three-way epistatic interaction among the first three variants was discovered through GMDR analysis. High-PRS individuals (scores 5–6) showed a 49-fold higher odds ratio of DM-DR compared to low-PRS individuals (scores 0–2; *p* < 0.0001). MAGMA analysis revealed enrichment in pathways related to protein degradation, vascular function, and neuronal signaling, with predominant upregulation in brain tissues. Significant PRS × lifestyle interactions were identified for fruit intake, coffee consumption, alcohol intake, eating duration, and physical activity, with lifestyle factors modifying genetic risk effects (all *p* < 0.003). **Conclusions:** These findings identify novel genetic variants and epistatic interactions in DM-DR pathogenesis, supporting the use of PRS-based risk stratification for intensive monitoring and personalized lifestyle interventions. The discovery of brain tissue-enriched pathways suggests DM-DR shares mechanisms with neurodegenerative diseases, expanding therapeutic targets beyond traditional vascular approaches.

## 1. Introduction

Diabetic retinopathy is one of the microvascular complications of diabetes, with recent advances in polygenic risk scores (PRS) providing new insights into genetic susceptibility and gene–lifestyle interactions in diabetic complications [1,2]. Characterized by progressive damage to the retinal blood vessels, diabetic retinopathy poses a significant risk of vision impairment and blindness if left untreated [3] and represents a leading cause of vision loss worldwide. The risk factors for diabetic retinopathy include diabetes duration, poor blood glucose control, hypertension, dyslipidemia, obesity, smoking, ethnicity, and genetics [4]. A complex interplay of genetic and environmental factors influences the development and progression of diabetic retinopathy. Understanding this interaction is crucial for targeted prevention and treatment strategies [4].

The pathogenesis of diabetic retinopathy involves dysregulation of several biological pathways, with numerous candidate genes implicated in its development and progression. These include genes involved in glucose metabolism [aldo-keto reductase family 1 member B1 (*AKR1B1*), *glucose transporter-1* (*GLUT1*)], angiogenesis [vascular endothelial growth factor (*VEGF*), *VEGF-A*], inflammatory processes [tumor necrosis factor-α (*TNF-α*), *interleukin (IL)-6*], oxidative stress responses [superoxide-dismutase-2 (*SOD2*), glutathione peroxidase-1 (*GPX1*)], and extracellular matrix remodeling [collagen type XVIII alpha 1 chain (*COL18A1*), *matrix metalloproteinase (MMP)-9*] [5]. These biological pathways have been increasingly recognized as critical therapeutic targets for treating broad retinal microvascular complications, with emerging evidence supporting multi-pathway approaches that simultaneously target angiogenesis, inflammation, and oxidative stress responses to achieve more comprehensive therapeutic efficacy [6].

The genetic architecture of diabetic retinopathy is characterized by both risk and protective variants. Notably, variants in genes such as nucleoside diphosphate kinase 3 (*NME3*), *LOC728699*, and fas-activated serine/threonine kinase (*FASTK*) have been associated with reduced susceptibility to diabetic retinopathy in certain populations [7]. Advances in genome-wide association studies (GWAS) have significantly expanded the understanding of diabetic retinopathy’s genetics, revealing multiple single-nucleotide polymorphisms (SNPs) associated with the disease and confirming its polygenic nature [8]. However, the intricate interplay between these genetic variants and their cumulative impact on the association with diabetic retinopathy remains incompletely understood. Furthermore, most genetic studies to date have been hampered by limited sample sizes and inadequate representation of diverse ethnic populations.

Furthermore, the influence of lifestyle factors on the genetic predisposition to diabetic retinopathy is poorly understood. Unhealthy lifestyles such as smoking, poor diet, alcohol consumption, and inadequate physical activity have been implicated in the pathogenesis of diabetic retinopathy [9]. Smoking has been consistently associated with an increased risk of diabetic retinopathy, while snacking and fast-food intake may exacerbate postprandial hyperglycemia and contribute to the development of diabetic retinopathy [10]. Conversely, moderate alcohol consumption, regular exercise, and slower eating duration have been linked to a reduced risk of diabetic retinopathy or improved glycemic control [11]. Despite recognizing these lifestyle factors as potential modulators of diabetic retinopathy risk, their interaction with polygenic variants has not been extensively studied.

The primary objective of this study was to identify polygenic variants associated with diabetic retinopathy through case–control analysis comparing patients with diabetic retinopathy (DM-DR) to diabetic patients without retinopathy (DM-NR) and to examine the interactions between PRS and modifiable lifestyle factors in the pathogenesis of DM-DR within an Asian population. Non-diabetic controls (ND) served as a reference group for demographic and clinical characterization but were excluded from genetic association analyses and PRS calculations, which were performed exclusively within the diabetic cohort to ensure phenotypic homogeneity. We evaluated interactions between PRS-defined genetic association strata and specific dietary exposures, including fruit and coffee consumption, to determine their combined effects on the susceptibility of diabetic retinopathy. This gene–environment interaction framework was designed to elucidate how genetic predisposition interacts with environmental factors, potentially informing precision medicine approaches for preventing diabetic retinopathy in high-risk diabetic populations.

## 2. Methods

### 2.1. Study Design and Data Source

This study utilized existing data from the Korea Genome Epidemiology Study (KoGES), which aims to establish the scientific foundation for personalized and preventive medicine by identifying risk factors associated with prevalent chronic diseases in the Korean population. The original KoGES data collection was conducted between 2010 and 2014, with individuals aged 40–79 years voluntarily enrolled in designated hospitals in major cities (n = 58,701) [12]. Our research team accessed the pre-existing KoGES dataset, which included phenotypic data and quality-controlled genotypic data. We did not participate in the original data collection, biological sample processing, or SNP genotyping procedures. Our contribution involved diagnosing diabetic retinopathy, conducting GWAS analysis for diabetic retinopathy, identifying novel genetic variants, developing polygenic risk scores, and performing statistical analysis of genetic associations with diabetic retinopathy using the available KoGES genotypic and phenotypic data.

### 2.2. Participants and Sample Size Calculation

From the KoGES dataset, participants with type 2 diabetes mellitus (T2DM) were identified based on fasting glucose levels ≥ 126 mg/dL, HbA1c levels ≥ 6.5%, or the use of antidiabetic medication [13]. The DM-DR group (n = 165) comprised patients recruited from St. Vincent Hospital ophthalmology department with diabetic retinopathy confirmed through: (1) dilated fundus examination by trained ophthalmologists, and/or (2) fundus photography interpretation, according to the International Clinical Diabetic Retinopathy Disease Severity Scale criteria. Participants were excluded if they had the following: (1) non-diabetic retinopathy eye diseases (cataracts, glaucoma; n = 2519), (2) incomplete genetic data, or (3) missing lifestyle questionnaire responses (Appendix A).

We conducted a priori power analysis using G*Power (version 3.1, University of Düsseldorf, Düsseldorf, Germany). Based on an assumed effect size of Cohen’s f^2^ = 0.15 (moderate), an alpha level of 0.05, and a desired power of 0.99, a minimum of 158 cases were required for an analysis of covariance (ANCOVA) model with up to 13 predictors, including covariates. Therefore, 165 DM-DR cases satisfied the power requirements for detecting gene–lifestyle interaction effects, rather than being based on the prevalence of diabetic retinopathy. The final study included 51,309 non-diabetic, 4873 DM-NR, and 165 DM-DR participants. Approval for all procedures involving human participation was obtained from the Institutional Review Boards of the Korea National Institute of Health for the KoGES (KBP-2015-055), St. Vincent Hospital (VIRB-20200128-020), and Hoseo University (1041231-150811-HR-034-01), adhering to the principles outlined in the Declaration of Helsinki. Written informed consent was obtained from all participants before their inclusion in the study.

### 2.3. Demographic and Biochemical Characteristics

Demographic and lifestyle parameters were assessed through standardized questionnaires via self-report. Residence was classified as urban or rural based on participants’ primary domicile for ≥6 months preceding study enrollment. Educational attainment was stratified into three categories: middle school, high school, or tertiary education (college degree or higher). Smoking status was operationalized according to established criteria [10], with participants classified as current smokers (≥20 cigarettes within the previous 6 months), former smokers, or never-smokers [14].

Participants wore light gowns and were barefoot when measuring body weight, height, waist circumference, and hip circumference using a Digital Scale and ergonomic circumference measuring tape (SECA, Los Angeles, CA, USA) [15]. Following a fasting period of at least 12 h, blood samples were collected in heparin-treated tubes. Hemoglobin A1c (HbA1c) levels were determined using an automatic analyzer (ZEUS 9.9; Takeda, Tokyo, Japan). Lipid profiles and serum creatinine concentration were analyzed in plasma, while glucose and creatinine concentrations were measured in serum using a Hitachi 7600 Automatic Analyzer (Hitachi LTD., Tokyo, Japan). Serum LDL cholesterol concentration was calculated with the Friedmann equation. Blood pressure was measured by a physician using a sphygmomanometer while participants were in a seated position, with three readings taken after a period of rest. Renal function was estimated using the estimated glomerular filtration rate (eGFR) equation generated by the Modification of Diet in Renal Disease (MDRD) study: 186 X serum creatinine concentration–1.154 X age–0.203 X 0.742 for women and 1 for men [16].

Based on 2005 revised National Cholesterol Education Program—Adult Treatment Panel III criteria adopted with Asians [17] and the Korean Society for the Study of Obesity criteria [18] for waist circumference, metabolic syndrome was defined as the presence of three or more of the following: (1) abdominal obesity (waist circumference ≥ 90 cm and 85 for men and women); (2) low high-density lipoprotein (HDL)-cholesterol level (<40 and 50 mg/dL for men and women, respectively); (3) elevated serum triglyceride level (≥150 mg/dL) or current anti-dyslipidemic medication usage; (4) elevated fasting blood glucose level (≥100 mg/dL) or current antidiabetic medication usage; and (5) elevated blood pressure (average systolic blood pressure (SBP) ≥ 130 mmHg or diastolic blood pressure (DBP) ≥ 85 mmHg) or current blood pressure medication usage.

### 2.4. Diagnosis of Diabetic Retinopathy

An ophthalmologist diagnosed diabetic retinopathy by a comprehensive fundus examination after pharmacologic pupil dilatation, and by fundus photography (TRC-NW65; Topcon, Tokyo, Japan) at a field angle of 45°. Diabetic retinopathy was diagnosed in the presence of any characteristic lesion determined by the Early Treatment for Diabetic Retinopathy Study (ETDRS) severity scale, a 12-step scale to document the severity of diabetic retinopathy using 7 field stereoscopic color fundus photographs, which included microaneurysms, hemorrhages, hard exudates, cotton wool spots, intraretinal microvascular abnormalities, venous beading, and retinal new vessels [19]. A diagnosis of diabetic retinopathy was given even if only one of the two eyes showed clinical signs of diabetic retinopathy. Examples of the fundus images are provided in Appendix A.

### 2.5. Assessment of Nutrient Intake and Diet Characteristics

Dietary intake was assessed by trained technicians using a validated 106-item semi-quantitative food frequency questionnaire (SQFFQ) to capture habitual food consumption patterns during the 12 months preceding baseline examination. The SQFFQ was previously validated against 3-day food records with acceptable correlation coefficients for nutrient intake estimation in Korean populations [16]. Daily intake for each food item was computed by multiplying the median consumption frequency by the corresponding portion size. Nutritional composition analysis was performed using CAN-Pro 2.0 software (Korean Nutrition Society, Seoul, Republic of Korea). Detailed methodology is provided in Appendix A.

Daily alcohol and coffee consumption was quantified based on self-reported frequency and quantity per consumption occasion. Alcohol intake was categorized as abstainers/light consumers (<20 g/day) or moderate consumers (≥20 g/day). Coffee consumption was similarly categorized into low (<0.5 cup/day) and high (≥3 g/day) intake groups. Regular physical activity was defined as moderate-intensity exercise (brisk walking, water aerobics, cycling, dancing, doubles tennis, power lawn mowing, hiking, or rollerblading) for ≥30 min per session, ≥3 days weekly.

### 2.6. Genotyping and Quality Control

Genomic DNA extracted from the blood samples of the Korean participants was processed using the Korean Chip (Affymetrix, Santa Clara, CA, USA) at the Korea Centers for Disease Control and Prevention (KCDC) [20]. To ensure the reliability of the genotyping data, the accuracy of genotyping was assessed using the Bayesian robust linear model with the Mahalanobis distance classifier (BRLMM) genotyping algorithm [21]. SNPs were deemed acceptable if they met specific criteria, including a genotyping accuracy of ≥98%, heterozygosity of <30%, missing genotype call rate of <4%, Hardy–Weinberg equilibrium (HWE) with *p* > 0.05, and no evidence of gender bias. Subsequently, KCDC provided the SNP data of the KoGES participants for the study (Figure 1).

### 2.7. Selection of SNPs for Diabetic Retinopathy Association by GWAS

GWAS was performed between DM-NR and DM-DR groups, adjusting for age, gender, BMI, diabetes duration, medication use, hypertension, eGFR, serum triglycerides, alcohol consumption, smoking status, and physical activity [22]. Population stratification was assessed by calculating the genomic inflation factor (λ) from the GWAS test statistics and examining quantile–quantile (Q-Q) plots of the observed versus expected *p*-values. All participants were confirmed to be of Korean ancestry based on self-reported ethnicity. The genomic inflation factor was λ = 1.05, indicating minimal population stratification within the Korean cohort, which is consistent with the homogeneous ancestry of the study population.

SNPs with Hardy–Weinberg equilibrium *p* > 0.05 and minor allele frequency < 0.01 were excluded. Initially, 1048 genetic variants were selected at a genome-wide significance threshold of *p* < 5 × 10^−7^, which represents a Bonferroni-corrected significance level (α = 0.05) accounting for multiple testing across the genome-wide SNP array between DM-NR and DM-DR. Subsequently, 282 unique gene names were identified from 884 of the 1084 SNPs using g:Profiler (https://biit.cs.ut.ee/gprofiler/snpense accessed on 13 November 2024). From 884 genetic variants, linkage disequilibrium-based pruning using R software 4.4.1 (r^2^ < 0.8) retained 103 independent SNPs. Twenty-six candidates were prioritized through a systematic literature review and Human Genome Epidemiology Navigator database, with 10 variants selected for epistatic analysis using the GeneMANIA (https://genemania.org/ accessed on 20 November 2024) prediction server.

### 2.8. PRS Construction with SNP-SNP Interaction and Validation

Epistatic interactions among the 10 selected SNPs were evaluated using generalized multifactor dimensionality reduction (GMDR), a non-parametric, model-free approach specifically designed to detect non-linear gene–gene interactions [23]. The analysis incorporated two covariate adjustment strategies: covariate set 1 (age, gender, BMI, diabetes duration) and covariate set 2 (covariate set 1 plus medication use, hypertension, eGFR, serum triglycerides, alcohol consumption, smoking status, and physical activity) [24]. Multiple PRS models incorporating 1 to 10 SNPs were systematically constructed and evaluated using rigorous selection criteria, including cross-validation consistency (10/10), balanced accuracy (*p* < 0.001), and statistical significance in GMDR analysis.

The final 3-SNP model, comprising *ABCA4*_rs17110929, *MMP2-AS1*_rs2576531, and FOXP1_rs557869288, was selected based on genome-wide significance, strongest epistatic effects, clinical applicability, and model parsimony. An unweighted count-based PRS approach was employed, summing risk alleles across the three selected SNPs, because external effect size estimates for epistatic interactions are not available, and this approach maintains clinical simplicity [22]. The PRS calculation followed: PRS = Σ(number of risk alleles across three SNPs), where each SNP contributes 0, 1, or 2 risk alleles per individual, generating scores ranging from 0 to 6. Participants were stratified into risk tertiles based on sample distribution and statistical power considerations: low-risk (0–2 alleles, n = 3880), medium-risk (3–4 alleles, n = 1082), and high-risk (5–6 alleles, n = 192). Internal validation using 10-fold cross-validation demonstrated perfect model consistency and addressed potential overfitting concerns given the modest DM-DR sample size (n = 165). Detailed PRS construction methodology is provided in Appendix A.

### 2.9. Multi-Marker Analysis of GenoMic Annotation (MAGMA) and Pathway Analysis

Gene-level and pathway analyses were performed using MAGMA v1.10 with the 1000 Genomes East Asian reference panel. Pathway enrichment analysis used curated gene sets from MSigDB databases with FDR correction (q < 0.05).

### 2.10. Sensitivity Analysis Based on Diabetic Duration

To assess the robustness of PRS and DM-DR associations, we conducted sensitivity analyses using three cohorts with progressive diabetic duration restrictions: (1) the full cohort without exclusions, (2) diabetes duration ≤ 20 years, and (3) diabetes duration ≤ 10 years. Logistic regression models with consistent covariate adjustment were applied to each cohort, calculating odds ratio (OR) and 95% confidence interval (CI) for PRS categories (low, medium, and high) with low-PRS as the reference. Heterogeneity across cohorts was assessed using Cochran’s Q test (*p* > 0.05 indicating no significant heterogeneity) and I^2^ statistics (<25% indicating low heterogeneity).

### 2.11. Statistical Analyses

Statistical analyses were conducted using PLINK version 2.0 (https://zzz.bwh.harvard.edu/plink/; accessed on 7 October 2024) and SAS (version 9.3; SAS Institute, Cary, NC, USA). We determined the normality of data distribution by univariate analysis. The skewness and kurtosis values of the parameters used in the study ranged between −0.1 and 0.1, indicating approximate normality. Covariate sets 1 and 2. Adjusted means and standard deviations for continuous variables were calculated across ND, DM-NR, and DM-DR groups using one-way ANCOVA, with covariate set 2. Multiple group comparisons were performed using Tukey’s post hoc test. The distribution of categorical variables across ND, DM-NR, and DM-DR groups was examined using the Chi-squared test.

Adjusted OR and 95% CI for the PRS influencing DM-DR were analyzed using logistic regression models with adjusted covariate sets 1 and 2. Low-PRS served as the reference category. To evaluate whether diabetes duration influenced the observed association between PRS and DM-DR, we conducted a sensitivity analysis by stratifying participants based on diabetes duration. Separate logistic regression models adjusting for covariate set 2 were applied to subsets excluding individuals with >10 years and >20 years of diabetic duration at the time of diabetic retinopathy assessment. Logistic regression with full samples (no exclusion) was also conducted. Adjusted OR and 95% CI were estimated for PRS group comparisons within each subgroup.

To investigate potential PRS–lifestyle interactions, we employed a two-way ANCOVA model. The model incorporated two main effects (PRS and lifestyle parameters), their interaction term (PRS × lifestyle), and covariate set 2. The general form of the two-way ANCOVA model with covariate set 2 was expressed as follows: DM-DR status = β_0_ + β_1_(PRS) + β_2_(lifestyle parameter) + β_3_(PRS × lifestyle parameter) + β_4−n_(Covariates) + ε. We dichotomized participants into higher or lower lifestyle groups using the 66th percentile of each variable as the threshold. Bonferroni correction was applied to account for multiple lifestyle parameter testing: Since 15 lifestyle parameters were evaluated for PRS interactions, the significance threshold was adjusted to *p* < 0.003 (α = 0.05/15) using Bonferroni correction for multiple comparisons. Interaction significance was evaluated using F-tests with this corrected threshold. Additionally, to quantify effect sizes and obtain adjusted OR and 95% CI for significant interactions, we performed logistic regression models including the main effects of lifestyle and PRS, their interaction term, and covariate set 2.

## 3. Results

### 3.1. Characteristics of the DM-DR Group

The mean age of the individuals in each group was as follows: ND (53.5 ± 0.03 years), DM-NR (57.4 ± 0.11 years), and DM-DR (58.2 ± 0.61 years). The duration of diabetes was markedly longer in the DM-DR group (17.2 ± 0.30 years) compared to the DM-NR group (4.23 ± 0.07 years). The proportion of males was the highest in the DM-DR (57.0%) among the three groups (ND: 32.9%; DM-NR: 49.8%) (Table 1).

The prevalence of current smokers was highest in the DM-DR group (7.8%), compared to 4.49% in the DM-NR group and 3.36% in the ND group (*p* < 0.001). Alcohol consumption was high across all three groups, with the highest prevalence in the DM-DR group (30.3%) compared to 8.0% and 4.36% in the DM-NR and ND groups, respectively (*p* < 0.001). Daily coffee consumption was highest in the DM-DR group (1.80 ± 0.114 cups/day). Exercise patterns showed significant differences across groups (*p* < 0.001), with a striking pattern in the DM-DR group where 89.1% exercised for 30–60 min, higher compared to the other groups. Fruit consumption was highest in the DM-DR group (>2.5 servings/day, 45.5%) and DM-NR (0.5–2.5 servings/day, 40.7%) compared to the other groups (*p* < 0.001). Fast food consumption (once a day and 2–3 times a week) was highest in the DM-DR (22.4 and 45.5%) compared to the ND group. Eating duration patterns also differed significantly (*p* < 0.001), with a higher proportion of slow eaters in the DM-DR group (20.0% taking 20–60 min) compared to the other groups (approximately 9%).

### 3.2. Association Between Metabolic Syndrome and DM-DR Group

Table 2 presents the OR comparing ND vs. DM-DR and DM-NR vs. DM-DR groups across various metabolic parameters without and with covariate adjustment. In unadjusted with covariates, DM-DR showed significantly higher odds compared to ND for metabolic syndrome (OR: 8.901, 95% CI: 5.904–13.42), low HDL cholesterol (OR: 5.063, 95% CI: 3.666–6.992), elevated SBP (OR: 5.419, 95% CI: 3.863–7.603), hypertension (OR: 1.756, 95% CI: 1.129–2.732), and reduced eGFR (OR: 2.523, 95% CI: 2.231–2.853). When comparing DM-NR vs. DM-DR, significant associations were observed for low HDL cholesterol (OR: 3.173, 95% CI: 2.287–4.402), elevated SBP (OR: 3.090, 95% CI: 2.194–4.353), and lower BMI (OR: 0.587, 95% CI: 0.423–0.816) and triglycerides (OR: 0.460, 95% CI: 0.315–0.671).

After adjusting for covariates, DM-DR maintained significantly higher odds compared to ND for metabolic syndrome (OR: 28.39, 95% CI: 13.80–58.39), low HDL cholesterol (OR: 25.68, 95% CI: 6.347–103.9), elevated SBP (OR: 15.61, 95% CI: 3.708–65.75) and DBP (OR: 9.115, 95% CI: 1.570–52.93), hypertension (OR: 10.09, 95% CI: 1.735–58.65), and reduced eGFR (OR: 1.705, 95% CI: 1.490–1.950). In DM-NR vs. DM-DR comparisons, significant associations persisted for low HDL cholesterol (OR: 3.342, 95% CI: 2.090–5.346), elevated SBP (OR: 4.349, 95% CI: 2.727–6.938) and DBP (OR: 5.698, 95% CI: 2.731–11.89), and hypertension (OR: 5.910, 95% CI: 2.815–12.41). Overall, DM-DR was consistently associated with adverse metabolic profiles, particularly dyslipidemia and hypertension, compared to both ND controls and DM-NR participants.

### 3.3. Selection of Genetic Variants Associated with DM-DR and Generation of PRS

The overall genetic variant associations with DM-DR based on the DM-NR are shown as a Manhattan plot (Appendix A), representing the statistical associations of genetic variants across chromosomes. The Q–Q plot exhibited the quantile distribution of observed *p*-values for genetic variants versus their expected *p*-values. The lambda value (genome inflation factor) was 1.045, indicating no inflation in genetic variant associations for DM-DR in the GWAS (Appendix A).

The adjusted OR of forkhead box protein P1 (*FOXP1*)_rs557869288 and *HLA-DQB1*_rs9274247 were less than 1, indicating that the minor allele was associated with reduced odds of DM-DR, suggesting the major allele (reference) is associated with higher odds (Table 3). SNP locations were mainly intronic, while mitochondrial ribosomal protein S33 (*MRPS33*)_rs1533933 and dopamine D2 receptor (*DRD2*)_rs4936270 were located in the 5′-untranslated region (UTR). *MRPS33*_rs1533933 and *DRD2*_rs4936270 variants were involved in mitochondrial function and neurological pathways, representing potential novel associations with DM-DR.

Ten genetic variants were selected based on interaction analysis using GMDR. The MAF and HWE of selected genetic variants met the criteria (Table 3). Although rs573262 and rs557869288 did not reach *p* < 5 × 10^−8^, they met the *p*-value threshold (0.001) for the sign test between trained balanced accuracy and test balance accuracy and achieved 10/10 cross-validation consistency (CVC). Among ten PRS models, 3-SNP, 9-SNP, and 10-SNP met the CVC criteria and *p* < 0.05 threshold (Table 3). The 3-SNPs comprised adenosine triphosphate (ATP)-binding cassette subfamily A member 4 (*ABCA4*) rs17110929, *MMP2-AS1* rs2576531, and *FOXP1* rs557869288. Its PRS was calculated by summing the risk alleles of the three genetic variants.

### 3.4. PRS Association with DM-DR

Among diabetic patients, 42.4% (n = 70) of 165 DM-DR were in the high-PRS group, compared to only 2.7% (n = 132) of 4945 DM-NR. Conversely, 20.6% (n = 34) of DM-DR patients were in the low-PRS group versus 77.8% (n = 2846) of DM-NR. DM-DR prevalence was 0.88% (low), 5.93% (medium), and 34.7% (high) among participants with diabetes, demonstrating a strong dose–response association. The PRS histogram (Appendix A) showed approximate normality with a symmetric, unimodal distribution. However, given that PRS values represented discrete allele counts (range: 0–6), the distribution showed a visual approximation to the normal distribution curve.

In logistic regression analyses comparing DM-NR and DM-DR groups (Figure 2, top panel), PRS categories (low, medium, and high) for 3-SNP, 9-SNP, and 10-SNP models were evaluated. After adjusting covariate sets 1 and 2, individuals with high-PRS consistently showed the highest OR, ranging from approximately 16 to 50, compared to low-PRS (reference) (Figure 2). In DM-NR vs. DM-DR comparisons (bottom panel), high-PRS with 3-SNP individuals displayed the highest OR based on low-PRS reference in both covariate models (OR = 42.6 and 45.7). Medium-PRS showed moderately increased odds, with an OR around 7 compared to low-PRS (Figure 2). Among PRS models, 3-SNP-PRS showed the highest OR in the high-PRS category after adjusting for covariates 1 and 2, compared to 9-SNP-PRS (OR = 16.3 and 19.7) and 10-SNP-PRS (OR = 18.6 and 23.5) (Figure 2). Despite large confidence intervals, both covariate adjustment models revealed similar patterns, supporting the consistency of these associations.

The AUCs for the 3-SNP, 9-SNP, and 10-SNP PRS models were 0.832 (95% CI: 0.803–0.862), 0.847 (95% CI: 0.816–0.877), and 0.846 (95% CI: 0.815–0.878), respectively. Although the 9-SNP and 10-SNP models showed slightly higher AUCs compared to the 3-SNP model, the differences were not statistically significant based on overlapping confidence intervals (Appendix A). These results suggest that even a simplified 3-SNP model retains a strong discriminative ability for predicting diabetic retinopathy, though expanded SNP panels may offer marginal gains.

### 3.5. Sensitive Analysis: Robustness of PRS Associations

To address potential concerns regarding the unbalanced group sizes (high-PRS group: n = 192 vs. low-PRS group: n = 3880) and assess whether our findings could be biased by small sample size or sparse data in the high-risk group, we performed comprehensive sensitivity analyses by progressively restricting the study population based on diabetes duration. These analyses allowed us to evaluate the consistency of genetic risk associations across different sample sizes and demographic subgroups. The association between genetic risk and diabetic retinopathy remained consistently strong and statistically significant across all sensitivity analyses (Appendix A). In the full sample without exclusions (n = 165 DM-DR cases), the high-PRS group demonstrated a 50.39-fold increased risk compared to the low-PRS group (95% CI: 30.8–82.3). When restricting to individuals with diabetes duration ≤ 20 years (n = 112 DM-DR cases), the high-PRS group maintained a substantial 37.09-fold increased risk (95% CI: 21.0–65.4). Further restriction to those with diabetes duration ≤ 10 years (n = 54 DM-DR cases) yielded a 33.77-fold increased risk (95% CI: 15.6–73.3) for the high-PRS group, demonstrating that even with progressively smaller sample sizes, the genetic associations remained robust with tight confidence intervals.

The medium-PRS group similarly showed consistent associations across all subgroups, with odds ratios ranging from 7.26 to 8.87, maintaining statistical significance despite reduced sample sizes. Importantly, heterogeneity testing confirmed excellent consistency across all analyses, with no significant heterogeneity observed for either medium-PRS (Cochran’s Q *p* = 0.992; I^2^ = 0%) or high-PRS (Cochran’s Q *p* = 0.609; I^2^ = 0%) comparisons. The absence of heterogeneity and the maintenance of tight confidence intervals across different sample sizes indicate that our findings are not influenced by the initial unbalanced group distribution or sparse data in the high-risk category. These sensitivity analyses demonstrate that the genetic risk associations are robust and independent of diabetes duration, with effect sizes remaining consistently large even when sample sizes are substantially reduced. The findings suggest that higher genetic burden may contribute to earlier onset of diabetic retinopathy, and importantly, confirm that our results are not biased by the smaller size of the high-PRS group relative to the low-PRS reference group.

### 3.6. Tissue-Specific Expression Analysis of DM-DR-Associated Variants

Figure 3A presents differential gene expression (DEG) analysis across 54 tissue types from the Genotype-Tissue Expression (GTEx) v8 dataset, focusing on DM-DR-associated genetic variants. Brain-related tissues (particularly cortex, frontal cortex, hippocampus, anterior cingulate cortex, and caudate and putamen basal ganglia) highlighted in red and exhibited the most significant upregulated gene expression. The results suggested a potential association of brain tissues with the pathophysiology of diabetic retinopathy. In contrast, down-regulated genes and combined up- and down-regulated DEG showed lower significance across tissues (Figure 3A). Other tissues, including metabolic and vascular-related tissues, exhibited limited differential expression patterns.

In Figure 3B, *MRPS33*_rs1533933 showed significant differential expression in the brain cortex (*p* = 3.68 × 10^−5^), supporting the observed brain tissue associations with DM-DR. Additionally, *MRPS33* expression varied significantly in other metabolic and structural tissues, including the pancreas, stomach, esophagus, colon, skeletal muscle, tibial nerve, and minor salivary gland, indicating broader systemic expression patterns. These findings support the observation that genetic variants associated with DM-DR predominantly affect brain tissue expression patterns, while also indicating expression associations in metabolic and peripheral tissues related to disease susceptibility.

### 3.7. MAGMA Gene-Set Analysis

The key biological pathways associated with DM-DR emphasized the roles of protein degradation, vascular dynamics, and neuronal signaling (Table 4). Notably, the glucose-induced degradation deficient (GID) complex (Bonferroni corrected *p* = 3.01 × 10^−7^) highlighted the protein regulatory mechanisms that may influence disease progression. Smooth muscle contraction (Bonferroni corrected *p* = 8.77 × 10^−6^) suggested vascular dysfunction as a contributing factor. Additionally, semaphorin-4D (*SEMA4D*)-induced cell migration and growth cone collapse (Bonferroni corrected *p* = 0.00108) pointed to neuronal signaling and structural remodeling (Table 4). These findings align with prior differential expression analysis, which showed strong genetic associations in brain and vascular tissues. Furthermore, pathways related to iron metabolism, genomic stability, and telomere maintenance suggest additional molecular mechanisms that may contribute to disease susceptibility.

### 3.8. Gene–Lifestyle Interaction in DM-DR Association

There was an interaction between genetic susceptibility and 15 lifestyle factors in the progression from DM-NR to DM-DR with and without adjustment of covariate set 2 (Bonferroni corrected *p* < 0.005; Table 5 and Appendix A). Among the dietary factors, fruit, fast foods, coffee, and alcohol intake exhibited significant gene-nutrient interactions with the PRS in adjusted logistic regression (*p* = 0.0023, <0.0001, <0.0001, and <0.0001, respectively).

Individuals with high-PRS and low fruit intake (<1 serving/day) had a higher association of DM-DR (OR = 81.3) compared to the medium-PRS group in adjusted logistic regression (OR = 9.82; Table 5). However, the OR of high-PRS was substantially lower in high fruit intake (OR = 47.03) than in low fruit intake in adjusted logistic regression (OR = 81.32). In high fruit intake, DM-DR incidence was lower in the high-PRS than in low fruit intake (Figure 4A). The interaction effect size between high fruit intake and high PRS versus the reference group showed a protective association (OR = 0.378, 95% CI: 0.149–0.953, *p* < 0.034), demonstrating that high fruit consumption significantly attenuates genetic risk in individuals with high PRS susceptibility. By contrast, in high fast food, coffee, and alcohol intake, the DM-DR incidence among individuals with high PRS was higher than in those with low intake (Figure 4B–D). Consistent with these DM-DR incidence results, the interaction effect sizes between high fast food, coffee, and alcohol intake and high PRS, relative to the reference group, showed positive associations (OR = 30.8, 95% CI: 30.0–31.5, *p* < 0.001 for fast food; OR = 74.4, 95% CI: 73.3–75.6, *p* < 0.001 for coffee; and OR = 3.99, 95% CI: 1.05–10.9, *p* < 0.05 for alcohol).

In addition to dietary factors, lifestyle behaviors such as eating duration also showed significant gene–lifestyle interactions in unadjusted and adjusted two-way ANCOVA (Table 5 and Appendix A). OR was strongly and positively associated with DM-DR in both participants with fast- and slow-eating speed in unadjusted and adjusted logistic regression analysis. Furthermore, OR was higher in participants with short eating duration than those with long eating duration (OR = 56.14 and OR = 35.27, *p* < 0.0001). The prevalence of DM-DR was significantly higher in the high-PRS group with rapid eating habits than in those who ate more slowly. Physical activity also demonstrated gene–lifestyle interactions, but OR in the high-exercise group was higher than that in the low-exercise group in both unadjusted and adjusted analyses (OR = 55.77 and OR = 32.03, *p* < 0.001). The cross-sectional nature of our study limited the interpretation of these findings due to potential reverse causation. This implies a potential link between metabolic dysregulation and genetic predisposition.

The stratified analysis by PRS tertiles (Appendix A) revealed differential associations between dietary factors and DM-DR risk across genetic susceptibility strata. For fruit intake, the protective association with high consumption was most pronounced in the medium-PRS group, with an adjusted OR of 0.422 (95% CI: 0.173–0.954, *p* < 0.05) and unadjusted estimates of 0.474 (95% CI: 0.271–0.830, *p* < 0.01), while this association was attenuated in low- and high-PRS strata. Coffee consumption demonstrated a complex gene–environment interaction pattern. Low coffee intake (<0.5 cup/day) was most strongly associated with increased DM-DR risk in the high-PRS group (OR = 4.99, 95% CI not reported, *p* < 0.0001), with progressively weaker associations in the medium-PRS (OR = 3.13) and low-PRS groups. Conversely, high coffee intake showed elevated risk across most PRS strata, with the strongest effects observed in low- and medium-PRS groups, suggesting a potential threshold effect.

Alcohol consumption exhibited a distinct interaction profile, with high intake showing a non-significant association with DM-DR in the high-PRS group (OR = 1.586, 95% CI: 0.167–15.02, *p* > 0.05), contrasting with significant associations in lower PRS strata. This pattern suggests potential attenuation of genetic risk by alcohol consumption. These stratified analyses demonstrate significant heterogeneity in dietary factor associations across genetic risk strata, indicating potential gene–environment interactions that may inform personalized prevention strategies. However, the cross-sectional study design precludes causal inference, and replication in prospective cohorts with larger sample sizes is warranted to confirm these preliminary findings and establish temporal relationships.

## 4. Discussion

This study identified novel genetic variants and gene–lifestyle interactions significantly associated with diabetic retinopathy susceptibility in a well-defined Korean cohort, KoGES. Our analysis presents several key advances: (1) confirmation of the components of metabolic syndrome, particularly hypertension, as significant diabetic retinopathy risk factors; (2) identification of three novel genome-wide significant variants (*ABCA4*_rs17110929, *MMP2-AS1*_rs2576531, and *FOXP1*_rs557869288) with substantial effect sizes; (3) discovery of additional variants (*MRPS33*_rs1533933 and *DRD2*_rs4936270) suggesting novel pathogenic pathways involving mitochondrial dysfunction and neurological signaling; (4) demonstration that identified variants are predominantly expressed in brain tissues and enriched in pathways regulating protein degradation, vascular function, and neuronal signaling; and (5) identification of significant gene–lifestyle interactions that modify diabetic retinopathy risk, particularly in individuals with high-PRS.

**Clinical Risk Factors and Blood Pressure.** Our findings revealed significant associations between blood pressure and diabetic retinopathy, with both SBP and DBP elevated in DM-DR patients despite comparable glycemic control, suggesting that hypertension drives retinal complications independent of glucose management. While landmark trials (DCCT, UKPDS) established hyperglycemia as the primary microvascular risk factor [25], emerging evidence indicated blood pressure plays an equally critical role [26]. The KNHANES-VII corroborated our findings, showing significantly higher SBP in diabetic retinopathy [27]. Mechanistically, elevated blood pressure induces retinal capillary shear stress, endothelial dysfunction, and blood–retinal barrier breakdown [28], potentially acting as a “second hit” in genetically predisposed individuals to accelerate retinal inflammation and neurodegeneration. These findings emphasize that optimal diabetic retinopathy management requires intensive blood pressure control alongside glycemic management, particularly in patients with moderate HbA1c levels and high genetic susceptibility.

**Novel Genetic Variants and Pathophysiological Mechanisms.** Traditional diabetic retinopathy genetic risk factors, including *ZNF395*, *PLEKHG5*, *VCAM-1*, *HIF-1α*, *COL18A1*, *EPO*, and *VEGF*, predominantly target single pathways related to angiogenesis and inflammatory responses with modest effect sizes (typically OR 1.2–1.8) [29]. In contrast, our novel variants demonstrate significantly larger individual effects and, more importantly, exhibit synergistic interactions absent from established risk factors. The key distinction lies in the multi-pathway convergence of our identified variants. While established genetic factors operate through independent, primarily additive mechanisms, the three-way interaction among *ABCA4*, *MMP2-AS1*, and *FOXP1* creates a synergistic pathogenic network where oxidative stress activation (*ABCA4* dysfunction) triggers matrix degradation (*MMP2-AS1* activation), while simultaneously impairing compensatory angiogenic responses (*FOXP1* dysregulation). This creates a positive feedback loop that substantially amplifies diabetic retinopathy risk beyond the sum of individual variant effects.

We discovered a significant three-way interaction among *ABCA4*_rs17110929, *MMP2-AS1*_rs2576531, and *FOXP1*_rs557869288 that substantially increases diabetic retinopathy susceptibility through complementary mechanisms, affecting retinal homeostasis. We proposed a synergistic mechanism of *ABCA4/MMP2-AS1/FOXP1* in diabetic retinopathy (Figure 5). *ABCA4* has established roles in retinal pathology, with mutations causing Stargardt disease through impaired clearance of toxic retinal aldehydes, leading to lipofuscin accumulation and retinal degeneration [30]. In diabetic retinopathy, the mechanistic association of *ABCA4* dysfunction extends beyond classical photoreceptor pathology [31]. The rs17110929 variant impairs *ABCA4*’s phospholipid flippase function, specifically reducing transport of N-retinylidene-phosphatidylethanolamine across photoreceptor disc membranes. This dysfunction leads to the accumulation of bis-retinoid compounds (A2E and A2PE) in the retinal pigment epithelium [32], which act as photosensitizers, generating reactive oxygen species upon light exposure [33]. In the hyperglycemic environment of DM, advanced glycation end-products further compromise *ABCA4* function by cross-linking membrane proteins and increasing lipid peroxidation. The accumulated oxidized lipofuscin components serve as damage-associated molecular patterns, activating complement pathways through C3 and factor H, triggering sustained inflammatory cytokine release (IL-1β, TNF-α, IL-6) that creates a chronic pro-inflammatory microenvironment characteristic of diabetic retinopathy progression [34].

The rs2576531 variant is located within *MMP2-AS1*, a long noncoding RNA that plays a crucial regulatory role in MMP2 expression and activity. MMP2-AS1 functions through a well-characterized molecular mechanism involving competitive endogenous RNA (ceRNA) activity: MMP2-AS1 binds to miR-34c-5p, preventing this microRNA from targeting and inhibiting *MMP2* mRNA, thereby leading to increased MMP2 protein expression and enzymatic activity [PMID: 35342412]. In diabetic retinopathy, this regulatory axis becomes particularly relevant as the rs2576531 variant may enhance *MMP2-AS1* expression or stability, resulting in more effective sequestration of miR-34c-5p and consequently higher MMP2 levels.

The mechanistic association of elevated *MMP2*, mediated through *MMP2-AS1* upregulation, involves specific substrate targeting of retinal vascular basement membrane components [35]. Enhanced MMP2 activity increases enzymatic degradation of type IV collagen, laminin, and fibronectin—key structural components of retinal capillary basement membranes. This enhanced matrix degradation disrupts the critical pericyte–endothelial cell interactions mediated by platelet-derived growth factor-β and angiopoietin-1 signaling, leading to pericyte detachment and loss [36]. Simultaneously, MMP2 cleaves latent transforming growth factor (TGF)-β binding proteins, releasing active TGF-β that promotes further pericyte loss and fibrotic responses [37]. The basement membrane degradation compromises the integrity of tight junction proteins (claudin-5, occludin, ZO-1), resulting in blood–retinal barrier breakdown and vascular leakage [38]. In diabetic conditions, hyperglycemia creates a pathogenic environment that amplifies this *MMP2-AS1/miR-34c-5p/MMP2* regulatory axis. High glucose directly upregulates *MMP2-AS1* expression through oxidative stress-responsive pathways, while simultaneously reducing miR-34c-5p levels through epigenetic modifications. This dual effect maximizes MMP2 protein production and activity. Additionally, advanced glycation end-products and reactive oxygen species activate pro-MMP2 through peroxynitrite-mediated mechanisms, creating a feed-forward amplification loop that sustains blood–retinal barrier breakdown and vascular pathology [39]. The MMP2-AS1 regulatory mechanism provides a novel therapeutic target, as interventions could potentially modulate this lncRNA-microRNA-mRNA network to control MMP2-mediated vascular damage in diabetic retinopathy.

This figure illustrates the convergent molecular pathways underlying diabetic retinopathy pathogenesis through three-way gene interactions. ABCA4 dysfunction leads to the accumulation of toxic retinal aldehydes and an increase in oxidative stress (ROS), thereby creating a pro-inflammatory microenvironment. Simultaneously, FOXP1 dysregulation contributes additional oxidative stress through vascular endothelial dysfunction and impairs the negative regulation of pathological angiogenesis, leading to unopposed VEGF signaling and aberrant neovascularization. This dual oxidative stress environment, resulting from both ABCA4 and FOXP1 dysfunction, synergistically upregulates MRPS33_rs1533933-*AS1* expression, which functions as a competitive endogenous RNA by sequestering miR-34c-5p, thereby preventing microRNA-mediated inhibition of *MMP2* mRNA expression. The resulting increase in MMP2 protein levels and enzymatic activity leads to degradation of retinal capillary basement membrane components (type IV collagen, laminin, fibronectin), compromising blood–retinal barrier integrity. The degraded basement membrane, combined with FOXP1-mediated dysregulation of angiogenic repair mechanisms, provides a permissive environment for abnormal new blood vessel formation characteristic of proliferative diabetic retinopathy. This convergent mechanism creates a positive feedback loop where inadequate vascular repair maintains ischemic conditions, perpetuating oxidative stress from multiple sources, sustaining MMP2-AS1 upregulation, and causing further vascular compromise.

*FOXP1* regulates vascular endothelial function and *VEGF* expression [40]. As a negative regulator of pathological angiogenesis, *FOXP1* dysfunction would dysregulate angiogenic responses critical in diabetic retinopathy, leading to unopposed *VEGF* signaling and aberrant neovascularization characteristic of proliferative diabetic retinopathy (PDR) [41]. Additionally, FOXP1 dysfunction contributes to endothelial cell oxidative stress production, creating an additional source of reactive oxygen species in the retinal microenvironment. The functional interaction creates a synergistic pathogenic mechanism through convergent oxidative stress pathways. ABCA4 dysfunction generates oxidative stress through the accumulation of toxic retinal aldehydes and lipofuscin compounds, while FOXP1 dysfunction contributes additional oxidative stress through vascular endothelial dysfunction. This dual oxidative stress environment amplifies the upregulation of the MMP2-AS1/miR-34c-5p/MMP2 regulatory axis, leading to enhanced MMP2-mediated matrix degradation of retinal capillary basement membranes. Concurrently, FOXP1 dysfunction independently dysregulates compensatory angiogenic responses through VEGF/PEDF pathway disruption, preventing adequate vascular repair [42]. This creates a multi-pathway convergence where the following occur: (1) oxidative stress amplification: both ABCA4 and FOXP1 dysfunction contribute to ROS accumulation, synergistically driving MMP2-AS1 upregulation; (2) vascular barrier breakdown: enhanced MMP2 activity (via MMP2-AS1) degrades basement membranes; (3) aberrant repair responses: FOXP1 dysfunction prevents proper angiogenic repair, maintaining ischemic conditions. The result is a positive feedback loop where inadequate vascular repair maintains ischemic conditions, sustaining oxidative stress from multiple sources, which perpetuates MMP2-AS1 upregulation and subsequent MMP2 activation, leading to further vascular compromise. This mechanistic convergence explains the substantial increase in diabetic retinopathy risk demonstrated by our polygenic risk score compared to individual variant effects, as the three pathways amplify each other’s pathogenic effects rather than working independently.

Additionally, we identified two novel variants with significant neurobiological implications that interact with the vascular pathway. *MRPS33*_rs1533933 affects a mitochondrial ribosomal protein essential for the translation of respiratory chain components [28]. This provides the first direct genetic evidence implicating the mitochondrial translational machinery in DM-DR pathogenesis, suggesting that impaired mitochondrial proteostasis is a primary mechanism rather than a secondary consequence of hyperglycemia. The exceptionally high energy demands of retinal neurons make them particularly vulnerable to such mitochondrial dysfunction [30,43]. The *DRD2*_rs4936270 variant affects dopamine receptor D2 function, providing evidence for dopaminergic dysregulation in diabetic retinopathy pathogenesis. Dopamine regulates not only visual signaling but also neurovascular coupling and blood–retinal barrier integrity [44]. This variant likely impairs retinal blood flow autoregulation and exacerbates neuroinflammation, connecting vascular and neuronal aspects of diabetic retinopathy.

Our key finding is that these variants interact synergistically rather than acting independently. The three-way interaction creates a “multi-hit” pathogenic mechanism affecting vascular integrity, oxidative stress response, and compensatory repair mechanisms. The interaction effects indicate that patients carrying multiple risk variants may need more aggressive preventive strategies targeting multiple pathways simultaneously. Brain tissue enrichment patterns support the emerging concept that diabetic retinopathy shares mechanisms with neurodegenerative diseases, with differential expression analysis revealing predominant upregulation of these genetic variants in brain tissues and enrichment in pathways related to protein degradation, vascular integrity, and neuronal signaling [45]. This pattern supports the concept that diabetic retinopathy is not solely a microvascular disorder but rather a systemic neurovascular disease with shared pathophysiological mechanisms between the retina and brain, aligning with the emerging ‘retina as a window to the brain’ hypothesis [46]. These findings expand the conceptual framework of diabetic retinopathy pathogenesis beyond traditional microvascular paradigms [47], suggesting that effective diabetic retinopathy management may require comprehensive approaches targeting both vascular dysfunction and neurodegeneration rather than strategies focused solely on anti-VEGF therapies.

**Gene–Lifestyle Interactions with Direction and Magnitude of Risk Modification.** In the present study, two-way ANCOVA interaction analysis employed a factorial design incorporating PRS tertiles and lifestyle parameters as main effects, their interaction term (PRS × lifestyle), and covariate set 2 (age, sex, diabetes duration, HbA1c, blood pressure) as covariates. Participants were dichotomized into higher or lower levels of the lifestyles using the 66th percentile threshold for each variable. Interaction significance was evaluated using F-tests with Bonferroni correction (*p* < 0.003 for 15 multiple comparisons). Our stratified analysis by PRS tertiles revealed differential associations between dietary factors and diabetic retinopathy risk across genetic susceptibility strata, demonstrating significant gene–environment interactions with distinct directional patterns. For fruit intake, the protective association was most pronounced in the medium-PRS group, where high fruit consumption showed a significant protective effect (adjusted OR = 0.422, 95% CI: 0.173–0.954, *p* < 0.05; unadjusted OR = 0.474, 95% CI: 0.271–0.830, *p* < 0.01). This protective association was attenuated in both low- and high-PRS strata, suggesting an optimal genetic susceptibility window where antioxidant-rich foods provide maximum protective benefit against the development of diabetic retinopathy.

Coffee consumption demonstrated complex gene–environment interaction patterns with threshold effects. Low coffee intake (<0.5 cup/day) showed the strongest association with increased diabetic retinopathy risk in the high-PRS group (OR = 4.99, *p* < 0.0001), with progressively weaker associations in medium-PRS (OR = 3.13) and low-PRS groups. This represents a dose–response relationship where genetic susceptibility amplifies the detrimental effects of insufficient coffee consumption. Conversely, high coffee intake showed elevated risk across most PRS strata, with the strongest effects in low- and medium-PRS groups, indicating potential harmful effects of excessive consumption regardless of genetic background. Alcohol consumption exhibited a distinct interaction profile, with high intake showing non-significant associations with DM-DR in the high-PRS group (OR = 1.586, 95% CI: 0.167–15.02, *p* > 0.05), contrasting with significant associations in lower PRS strata. This pattern suggests potential attenuation of genetic risk by alcohol consumption, though the wide confidence intervals indicate substantial uncertainty in this estimate.

The observed gene–lifestyle interactions provide insights into the complex biological mechanisms underlying diabetic retinopathy pathogenesis. In genetically susceptible individuals (high-PRS), protective lifestyle factors may paradoxically increase disease risk through several interconnected pathways. Coffee consumption interactions likely involve the role of genetic variants in vascular regulation and oxidative stress responses [48]. While coffee’s bioactive compounds (caffeine, chlorogenic acids) typically modulate endothelial function beneficially [49], variants in *MMP2-AS1* and *ABCA4* may alter antioxidant processing, potentially converting protective effects into pro-inflammatory responses that exacerbate retinal microvascular damage. Physical activity interactions reflect the complex relationship between exercise-induced metabolic stress and genetic predisposition [50]. High-PRS individuals carrying variants in neuronal signaling pathways (*FOXP1*) may experience altered neurovascular coupling responses to exercise, leading to paradoxical retinal hypoxia or abnormal angiogenic signaling [42]. We found a contradictory finding that current exercisers showed a higher prevalence of DM-DR across all PRS categories than non-exercisers. While the pronounced PRS gradient among exercisers may indicate persistent genetic susceptibility, our single-time-point assessment could not determine whether current exercise patterns preceded disease onset, followed diagnosis, or represented ongoing management strategies. The cross-sectional nature prevents us from evaluating the temporal relationship between genetic susceptibility, lifestyle adoption, and disease development. These methodological limitations highlight the critical need for longitudinal studies with serial assessments to establish causal relationships and validate our findings. Dietary interactions may involve genetic variants affecting antioxidant metabolism. The protective effects of fruit-derived compounds may be diminished in high-PRS individuals due to altered cellular uptake mechanisms (*ABCA4* variants) or modified enzymatic processing of protective compounds.

**Clinical Translation and Precision Medicine Applications.** From a translational perspective, our findings provide a practical framework for incorporating genetic screening into routine diabetic care. The PRS model developed in this study identified individuals with up to a 53-fold association for developing diabetic retinopathy, offering clinicians a powerful risk stratification tool that extends beyond traditional clinical markers. In clinical practice, PRS testing could be implemented at diabetes diagnosis or early in disease management to guide several key clinical decisions. Screening intervals can be personalized based on genetic risk, with low-PRS patients safely undergoing standard annual retinal examinations as recommended by current guidelines [51], while high-PRS patients warrant intensified monitoring every 6 months or less, enabling early detection of retinal changes and timely intervention. This approach optimizes resource allocation while ensuring high-risk individuals receive appropriate surveillance.

PRS can inform the timing of therapeutic interventions, particularly panretinal photocoagulation (PRP). While the Early Treatment Diabetic Retinopathy Study (ETDRS) traditionally recommended PRP for PDR [52], recent practice has shifted toward earlier intervention in severe NPDR cases [53]. High-PRS patients with severe NPDR could be prioritized for early PRP, as their genetic susceptibility suggests a higher likelihood of progression to vision-threatening stages. This precision approach could prevent irreversible vision loss in individuals with a genetic predisposition. For patients with vitreous hemorrhage, PRS could supplement traditional criteria for early vitrectomy [54]. High-PRS patients may benefit from more aggressive surgical intervention given their genetic predisposition to poor outcomes, potentially improving long-term visual prognosis through earlier surgical management. The identified gene–lifestyle interactions provide actionable guidance for individualized prevention strategies. High-PRS patients can receive targeted counseling emphasizing specific dietary modifications, such as increasing fruit intake to >2 servings daily, which our data suggest reduces genetic risk by approximately 60%.

Beyond individual clinical applications, integrating PRS with lifestyle factors offers opportunities for population-based diabetic retinopathy prevention strategies. Public health programs can target genetically high-risk subgroups for diabetic retinopathy with customized interventions, such as community-based nutrition education emphasizing protective dietary patterns [55]. Health systems could allocate screening resources more efficiently by stratifying diabetic populations based on PRS, enabling targeted surveillance in resource-limited settings. These findings support a precision public health approach where genetic information informs both individual care and broader prevention strategies, ultimately reducing the societal burden and healthcare costs associated with diabetic retinopathy-related vision loss.

**Study Limitations.** First, the cross-sectional design fundamentally limits our ability to establish causal relationships, particularly for PRS–lifestyle interactions where reverse causation may explain observed associations. Physical activity displayed a paradoxical positive association with diabetic retinopathy risk, which likely reflects reverse causation where patients with established retinopathy increased their physical activity following medical advice or diabetes management interventions, rather than exercise being a risk factor. While some lifestyle factors demonstrated protective effects consistent with longitudinal studies—such as dietary patterns resembling the Mediterranean diet showing benefits in trials like ACCORD-EYE [56]—other factors, particularly physical activity, displayed patterns more consistent with disease-driven behavioral changes rather than genuine protective effects. Second, a significant limitation is that we did not distinguish between NPDR and PDR, although diabetic retinopathy is clinically heterogeneous. These represent clinically distinct stages with different pathophysiological mechanisms: NPDR primarily involves microvascular changes such as microaneurysms and hemorrhages, while PDR is characterized by retinal neovascularization and carries a higher risk of vision-threatening complications. Given that NPDR and PDR may be associated with different genetic profiles and progression pathways, the heterogeneity within our diabetic retinopathy cohort may have influenced our genetic association results. Additionally, we were unable to assess whether the identified genetic variants could predict progression from NPDR to PDR, which would have significant clinical implications for risk stratification and monitoring strategies. Third, the eye disease exclusion was necessary to isolate diabetic retinopathy-specific genetic associations, but may have introduced selection bias by removing individuals with multiple ocular conditions. Additionally, the different diagnostic approaches between groups may affect interpretation: the DM-NR group potentially included undiagnosed early-stage diabetic retinopathy cases due to the absence of ophthalmological screening, while the DM-DR group underwent comprehensive ophthalmological examination. This potential misclassification would bias results toward the null hypothesis, making our genetic associations more conservative estimates of true effect sizes. Fourth, our sensitivity analyses by diabetes duration showed consistent genetic associations across all subgroups (≤10 years, ≤20 years, full sample) with no heterogeneity, suggesting that temporal progression does not significantly confound our genetic risk estimates. Fifth, the substantial case–control imbalance between DM-DR cases (n = 165) and DM-NR controls (n = 4281) represents a critical methodological limitation with several consequences. This 26:1 ratio may lead to inflated effect size estimates through the Winner’s Curse phenomenon, reduced statistical power to detect true associations, and increased susceptibility to population stratification artifacts. For PRS construction, this imbalance critically affects the interpretability of our 49-fold risk increase, as the high-risk tertile comprises only 192 individuals compared to 3880 in the low-risk group, creating inherently unstable risk estimates. Power calculations indicate our sample provides 80% power to detect effect sizes (OR) ≥ 2.5 for variants with minor allele frequencies ≥ 0.1, but may be underpowered for smaller effects or rare variants. The observed diabetic retinopathy prevalence of 3.7%, while clinically realistic, means genetic effect sizes may be overestimated due to outcome rarity, and case–control imbalances of this magnitude are known to produce spurious associations that may explain why our effect sizes appear larger than those in balanced GWAS studies. Sixth, lifestyle behaviors were assessed through self-reported questionnaires with technician assistance, introducing several potential biases. Self-reported dietary intake is susceptible to recall bias, social desirability bias (over-reporting of healthy foods like fruits, under-reporting of alcohol), and portion size misestimation. The 12-month recall period for dietary habits may not accurately reflect long-term dietary patterns relevant to chronic disease development. Additionally, lifestyle factors were measured at a single time point, failing to capture behavioral changes over the course of diabetes progression that could influence the development of diabetic retinopathy. Lastly, our cohort was drawn exclusively from a Korean population, and genetic associations may differ across ethnicities due to population-specific allele frequencies and environmental exposures. Additional limitations include potential residual confounding from unmeasured factors such as glycemic variability, medication adherence, and socioeconomic status.

**Future Research Directions.** Given these limitations, our genetic findings require urgent validation in larger, more balanced cohorts before clinical implementation. The extreme effect sizes observed (49-fold PRS increase) should be interpreted with considerable caution until replicated in studies with adequate case numbers (n > 1000) and more balanced case–control ratios. Replication in established GWAS databases such as the UK Biobank, FinnGen, or other DM-DR consortia would strengthen the evidence for our identified variants. While sensitivity analysis excluding variants with minor allele frequencies < 0.05 yielded consistent results for *ABCA4*_rs17110929, *MMP2-AS1*_rs2576531, and *FOXP1*_rs557869288, supporting their robustness, validation in populations with larger DM-DR case numbers remains essential for clinical translation.

Prospective longitudinal studies with a minimum 10-year follow-up and balanced case–control ratios should establish temporal relationships between genetic susceptibility, lifestyle factors, and incidence of diabetic retinopathy. Multi-ethnic validation studies should replicate PRS performance across diverse populations with ethnic-specific calibration to ensure broader applicability. Randomized intervention trials should test whether PRS-guided lifestyle modifications reduce the incidence of diabetic retinopathy compared to standard care, providing evidence for personalized prevention strategies. Finally, biomarker studies should identify circulating metabolites that mediated PRS–environment interactions as targets for potential intervention. These research priorities will accelerate the translation of genetic risk prediction into evidence-based clinical practice for DM-DR prevention, while addressing the current limitations that restrict immediate clinical application of our findings.

## 5. Conclusions

This study advances our understanding of diabetic retinopathy pathogenesis by identifying novel genetic variants (*ABCA4*_rs17110929, *MMP2-AS1*_rs2576531, *FOXP1*_rs557869288) and additional loci (*MRPS33*_rs1533933, *DRD2*_rs4936270). Our findings reframe diabetic retinopathy as a complex disorder involving mitochondrial, neurovascular, and protein homeostasis pathways beyond traditional inflammatory mechanisms. We demonstrated significant PRS–lifestyle interactions, where individuals with high-PRS showed a substantially modified association of diabetic retinopathy in response to diet, alcohol, coffee consumption, and physical activity. The association of brain-tissue-enriched genetic pathways with retinal pathology supports the ‘retina as a window to the brain’ hypothesis. These results enable clinical translation through PRS-guided screening and personalized lifestyle counseling while emphasizing blood pressure control as a critical preventive measure. Future studies in diverse populations and about the functional validation of identified pathways remain essential for developing targeted therapeutic strategies.

## Figures and Tables

**Figure 1 nutrients-17-02983-f001:**
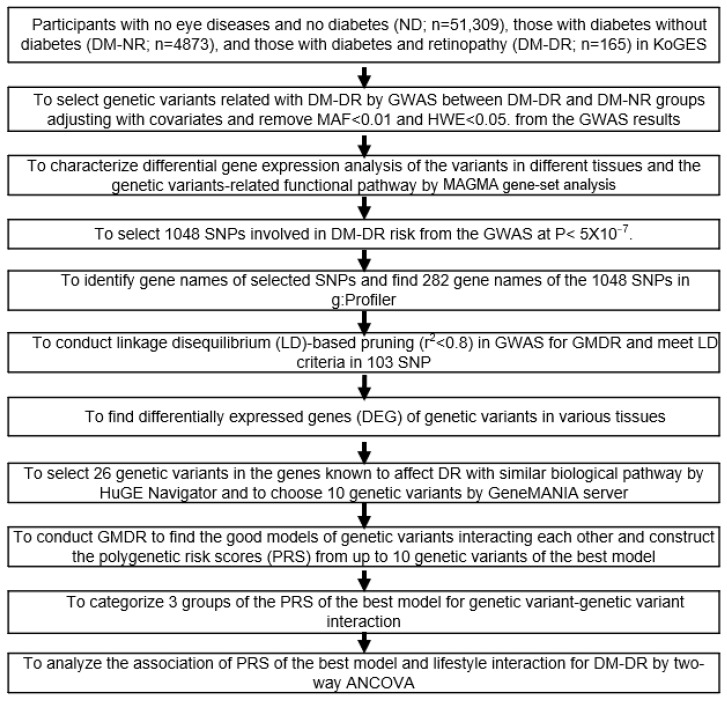
Flowchart for analyzing genetic and lifestyle interactions for diabetic retinopathy association in the Korea Genome Epidemiology Study (KoGES) cohort. GWAS analysis was conducted between diabetic patients without retinopathy (DM-NR) and diabetic patients with retinopathy (DM-DR) groups using two covariate adjustment models: Covariates 1 (red squares): age, gender, body mass index, and diabetic duration; Covariates 2 (blue squares): extended adjustment including residence area, waist and hip circumferences, hypertension status, lipid profiles, estimated glomerular filtration rate, smoking, alcohol intake, physical activity, and medications (lipid-lowering, antihypertensive, and antidiabetic). Abbreviations: SNP, single-nucleotide polymorphism; GWAS, genome-wide association study; MAF, minor allele frequencies; MAGMA, Multi-marker Analysis of GenoMic Annotation; HWE, Hardy–Weinberg equilibrium; GMDR, generalized multifactor dimensionality reduction; HuGE, Human Genome Epidemiology; ANCOVA, analysis of covariance.

**Figure 2 nutrients-17-02983-f002:**
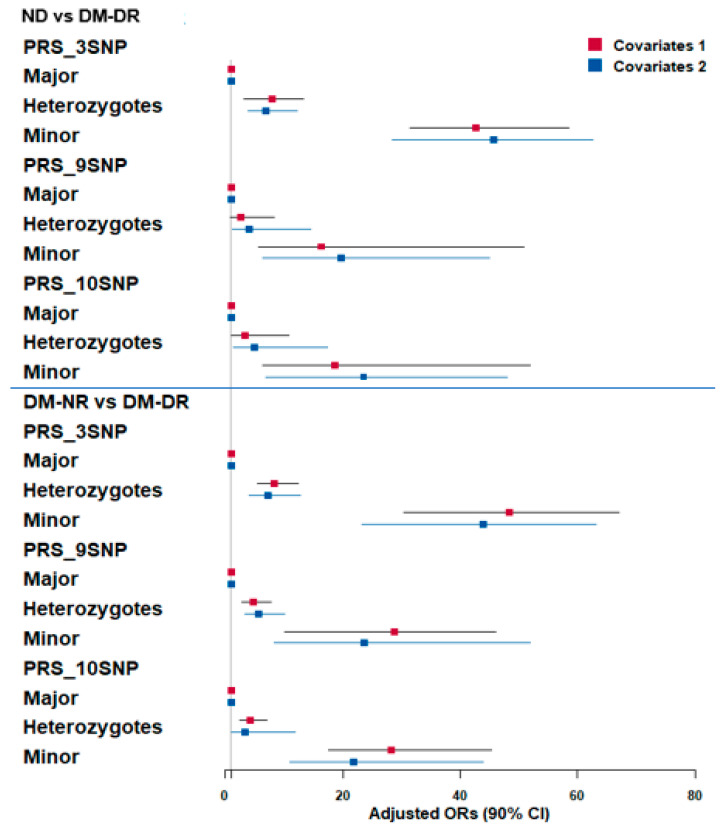
Adjusted odds ratios (OR) and 95% confidence intervals (CI) for polygenic risk score (PRS) associations with diabetic retinopathy risk. Three PRS models (3-, 9-, and 10-genetic variants) were constructed using GWAS and GMDR analysis. PRS stratification: Major = high-risk group (highest number of risk alleles); Minor = low-risk group (lowest number of risk alleles); Heterozygotes = intermediate risk. Upper panel: Non-diabetic controls (ND) vs. diabetic retinopathy patients (DM-DR) groups. Lower panel: Diabetic patients without retinopathy (DM-NR) vs. DM-DR groups. Color coding: red squares indicated adjusted OR in adjusted logistic regression with covariates 1 adjustment (age, gender, bodu mass index, diabetic duration); blue squares indicated adjusted OR in adjusted logistic regression with covariates 2 adjustment (extended model including residence area, waist/hip circumferences, hypertension, lipid profiles, estimated glomerular filtration rate, smoking, alcohol, physical activity, medications).

**Figure 3 nutrients-17-02983-f003:**
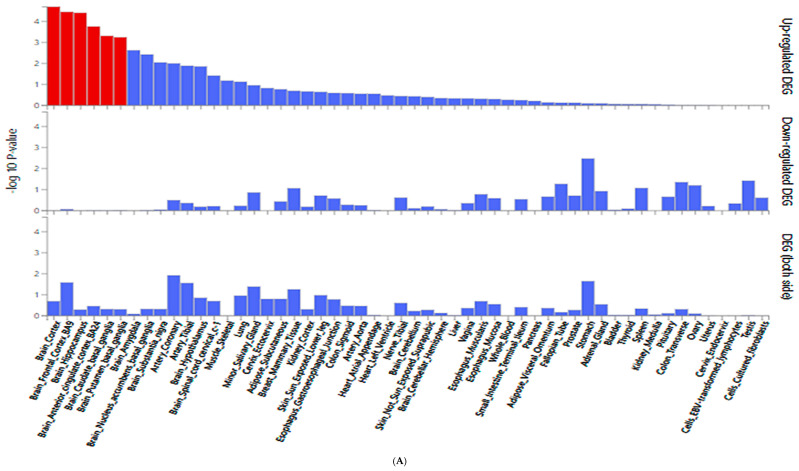
Gene expression analysis of genetic variants associated with diabetic retinopathy (DM-DR). (**A**) Differentially expressed genes (DEG) for SNPs identified from a genome-wide association study comparing patients with and without DM-DR. Red bars indicate significantly enriched DEG sets (Bonferroni corrected *p* < 0.05). (**B**) Expression quantitative trait loci (eQTL) analysis showing gene expression levels across non-risk, heterozygote, and risk allele carriers in different tissue types from the Genotype-Tissue Expression (GTEx) v8 dataset. The red *p*-value indicates significant differences in gene expression among allele groups (Bonferroni corrected *p* < 0.05). *MRPS33*, mitochondrial ribosomal protein S33.

**Figure 4 nutrients-17-02983-f004:**
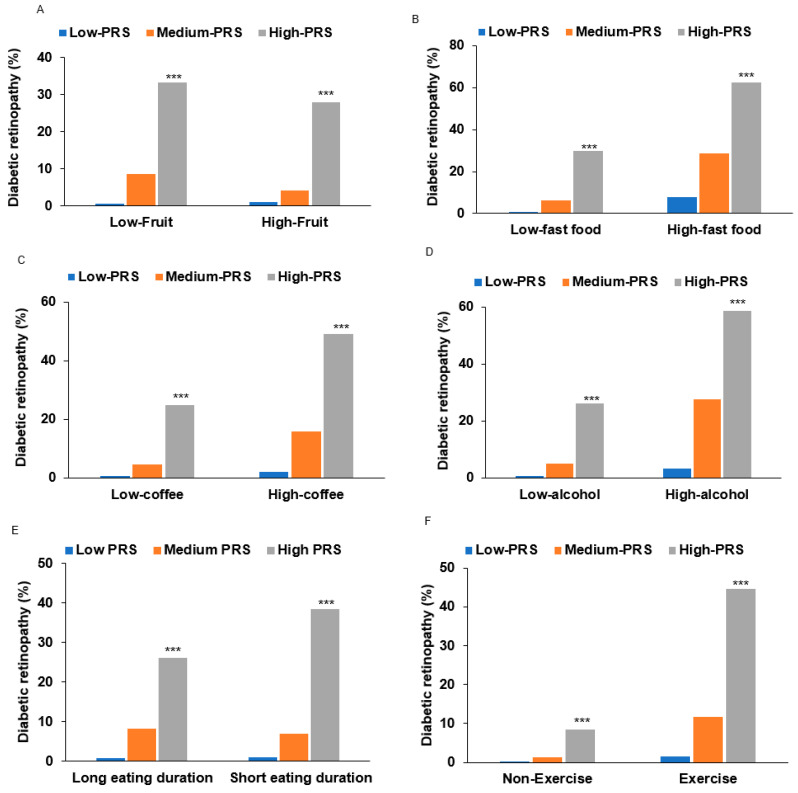
Diabetic retinopathy (DM-DR) incidence according to lifestyle factors interacting with polygenic risk scores (PRS). (**A**) DM-DR incidence according to fruit intake (cutoff: < 1 serving/day). (**B**) DM-DR incidence according to fast foods (cutoff: <1 serving/day). (**C**) DM-DR incidence according to coffee intake (cutoff: <0.5 cup/day). (**D**) DM-DR incidence according to alcohol intake (cutoff: <3 g/day). (**E**) DM-DR incidence according to eating duration (cutoff: <15 min/each meal). (**F**) DM-DR incidence according to physical exercise (cutoff: moderate-intensity exercise for 30 min/day). PRS composition: 3-SNP model comprising *ABCA4*_rs17110929, *MMP2-AS1*_rs2576531, and *FOXP1*_rs557869288, calculated by summing risk alleles. Statistical significance: Abbreviations: *ABCA4*, ATP-binding cassette subfamily A member 4; *MMP2-AS1*, matrix metalloproteinase-2; *FOXP1*, forkhead box protein P1. *** Significantly different among the PRS categories at *p* < 0.001 in each low and high intake in χ^2^ test.

**Figure 5 nutrients-17-02983-f005:**
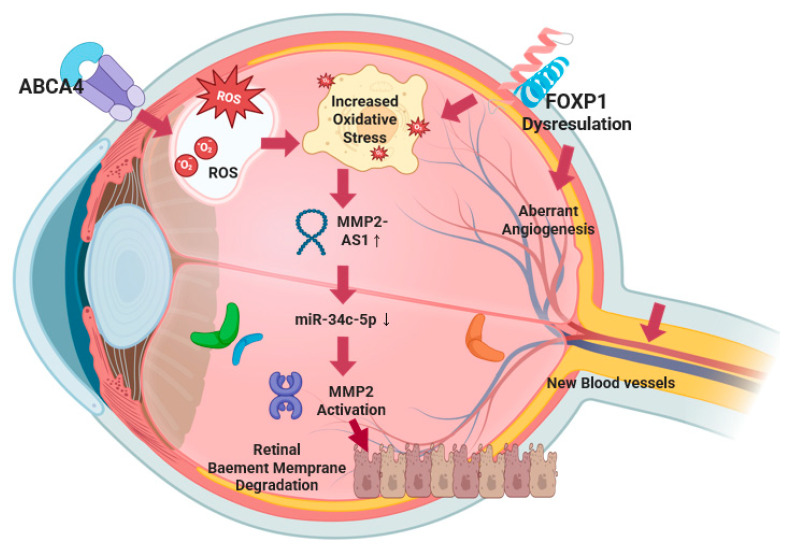
A synergistic mechanism of *ABCA4/MMP2-As1/FOXP1* in diabetic retinopathy.

**Table 1 nutrients-17-02983-t001:** Characteristics of diabetes without and with diabetic retinopathy.

	ND (n = 51,309)	DM-NR (n = 4873)	DM-DR (n = 165)	*p* Value
Age (year)	53.5 ± 0.03 ^b^	57.4 ± 0.11 ^a^	58.2 ± 0.61 ^a^	<0.001
Gender (Male, N, %)	16,881 (32.9)	2426 (49.8)	94 (57.0)	<0.001
Residence area (city, N, %)	19,705 (59.2)	2813 (57.7)	98 (59.4)	0.0521
Education (N, %) Middle school High school College+	6550 (18.8) 25,664 (73.7) 2598 (7.5)	1016 (26.7) 2571 (67.5) 220 (5.78)	50 (30.2) 106 (64.2) 9 (5.4)	<0.001
Height (cm)	160.7 ± 0.02 ^b^	160.7 ± 0.08 ^b^	162.0 ± 0.41 ^a^	0.0036
BMI (kg/m^2^)	23.8 ± 0.01 ^b^	25.1 ± 0.05 ^a^	25.5 ± 0.32 ^a^	<0.001
Diabetic duration (years)	0 ± 0 ^c^	4.23 ± 0.07 ^b^	17.2 ± 0.30 ^a^	<0.001
Former smoking (N, %)	2225 (4.61)	300 (6.16)	20 (12.1)	<0.001
Current smoking (N, %)	1619 (3.36)	219 (4.49)	13 (7.8)	
Alcohol (N, %)	2237 (4.36)	390 (8.0)	50 (30.3)	<0.001
Coffee (cup/day)	1.23 ± 0.005 ^b^	1.04 ± 0.018 ^c^	1.80 ± 0.114 ^a^	<0.001
Exercise (<30 min 30–60 min >60 min)	29,297 (57.1) 13,186 (25.7) 8825 (17.2)	2500 (51.3) 1301 (26.7) 1072 (22.0)	17 (10.3) 147 (89.1) 1 (0.61)	<0.001
Fruit intake (<0.5 servings/day) 0.5–2.5 servings/day >2.5 servings/day	17,137 (33.4) 21,498 (41.9) 12,673 (24.7)	1959 (40.2) 1983 (40.7) 931 (19.1)	48 (29.1) 42 (25.5) 75 (45.5)	<0.001
Fast foods (No 2–3 times/week 1/day)	41,201 (80.3) 9543 (18.6) 575 (1.12)	3864 (79.3) 940 (19.3) 67 (1.36)	113 (68.5) 37 (22.4) 15 (9.09)	<0.001
Eating duration (<10 min 10–20 min 20–60 min)	20,164 (39.3) 26,527 (51.7) 4628 (9.02)	1964 (40.3) 2461 (50.5) 502 (9.23)	58 (35.2) 74 (44.9) 33 (20.0)	<0.001

Data are presented as adjusted means ± standard errors from analysis of covariance (ANCOVA) for continuous variables, and as numbers and percentages for categorical variables analyzed by χ^2^ test. Covariates: age, gender, diabetic duration, body mass index (BMI), smoking, alcohol intake, physical activity, estimated glomerular filtration rate, hypertension, serum triglyceride, lipid-lowering, antihypertensive, and antidiabetic medications. ND, non-diabetics; DM-NR, diabetes without retinopathy; DM-DR, diabetic retinopathy. ^a,b,c^ Means on the same row with different superscript letters indicate a significant difference among the groups by the Tukey test at *p* < 0.05.

**Table 2 nutrients-17-02983-t002:** Odds ratio and 95% confidence intervals for metabolic syndrome (MetS) and its components without and with diabetic retinopathy.

	No Adjustment		Adjusted for Covariates ^1^	
	ND vs. DM-DR ^2^	DM-NR vs. DM-DR ^3^	ND vs. DM-DR ^2^	DM-NR vs. DM-DR ^3^
Height (cm)	1.132(0.734–1.746)	1.500 (0.964–2.335)	1.408(0.687–2.884) ^1^	1.574 (0.767–3.228) ^2^
MetS (N, %)	8.901(5.904–13.42)	0.879 (0.582–1.328)	28.39 (13.80–58.39)	1.292 (0.686–2.435)
BMI (kg/m^2^)	1.130 (0.817–1.563)	0.587 (0.423–0.816)	2.104 (0.560–7.912)	1.386 (0.875–2.197)
Total cholesterol (mg/dL)	0.117 (0.043–0.316)	0.161 (0.060–0.436)	0.181 (0.066–2.498)	1.347 (0.427–4.244)
HDL (mg/dL)	5.063 (3.666–6.992)	3.173 (2.287–4.402)	25.68 (6.347–103.9)	3.342 (2.090–5.346)
LDL (mg/dL)	0.207 (0.077–0.559)	0.305 (0.112–0.826)	0.390 (0.142–1.070)	0.803 (0.288–2.242)
eGFR (mL/min/1.73 m^2^)	2.523 (2.231–2.853)	0.993 (0.963–1.024)	1.705 (1.490–1.950)	0.994 (0.973–1.016)
TG (mg/dL)	0.860 (0.592–1.250)	0.460 (0.315–0.671)	1.461 (0.324–6.593)	0.888 (0.533–1.480)
SBP (mmHg)	5.419 (3.863–7.603)	3.090 (2.194–4.353)	15.61 (3.708–65.75)	4.349 (2.727–6.938)
DBP (mmHg)	1.637 (1.052–2.545)	1.397 (0.891–2.191)	9.115 (1.570–52.93)	5.698 (2.731–11.89)
Hypertension (N, %)	1.756 (1.129–2.732)	1.456 (0.928–2.284)	10.09 (1.735–58.65)	5.910 (2.815–12.41)

ND, non-diabetics; DM-NR, diabetes without retinopathy; DM-DR, diabetes with retinopathy. ^1^ Covariates: age, gender, diabetic duration, body mass index (BMI), smoking, alcohol intake, physical activity, hypertension, estimated glomerular filtration rate (eGFR), serum triglyceride, and lipid-lowering, antihypertensive, and antidiabetic medications in adjusted logistic regression. ^2^ Reference: ND in logistic regression. ^3^ Reference: DM-NR in logistic regression. TG, triglyceride; SBP, systolic blood pressure; DBP, diastolic blood pressure.

**Table 3 nutrients-17-02983-t003:** Characteristics of genetic variants that influence diabetic retinopathy risk.

^a^ CHR	^b^ SNP	Base Pair	^c^ A1	^d^ A2	^e^ OR (95% CI)	^f^ *p* Value	Gene Names	Location	^g^ MAF	^h^ HWE
1	rs573262	18656738	G	A	2.23 (1.66–3.01)	1.23 × 10^−7^	*IGSF21*	Intron	0.101	0.554
1	rs17110929	94510807	T	A	7.91 (5.71–10.9)	1.05 × 10^−35^	*ABCA4*	Intron	0.046	0.637
3	rs557869288	71316275	T	A	0.02 (0.003–0.14)	4.53 × 10^−7^	*FOXP1*	Nmd transcript	0.119	0.175
6	rs9274247	32631295	A	G	0.28 (0.21–0.38)	1.50 × 10^−15^	*HLA-DQB1*	Nmd transcript	0.404	0.626
7	rs1533933	140715025	G	C	2.94 (2.22–3.89)	4.51 × 10^−14^	*MRPS33*	5 prime UTR	0.099	0.394
11	rs4936270	113318408	T	C	3.26 (2.52–4.22)	2.20 × 10^−19^	*DRD2*	5 prime UTR	0.188	0.005
15	rs72712070	27132987	G	A	2.78 (2.03–3.80)	1.47 × 10^−10^	*GABRA5*	Intron	0.083	0.767
16	rs2576531	55482887	A	T	4.32 (3.25–5.74)	6.43 × 10^−24^	*MMP2* *-AS1*	Intron	0.105	0.896
16	rs733616	5511363	C	G	3.81 (3.05–4.75)	3.31 × 10^−32^	*RBFOX1*	Intron	0.253	0.103
17	rs56899958	3424113	G	A	6.11 (3.95–9.46)	4.62 × 10^−16^	*TRPV3*	Nmd transcript	0.025	0.148

^a^ Chromosome; ^b^ single nucleotide polymorphism; ^c^ minor allele; ^d^ major allele; ^e^ odds ratio and 9% confidence intervals; ^f^ statistical significance of diabetic retinopathy (DM-DR) group after adjusting for age, gender, diabetic duration, BMI, smoking, alcohol intake, estimated glomerular filtration rate, hypertension, physical activity, and lipid-lowering, antihypertensive, and antidiabetic medication; ^g^ Minor allele frequency; ^h^ Hardy–Weinberg equilibrium.

**Table 4 nutrients-17-02983-t004:** MAGMA gene-set analysis of diabetic retinopathy-associated genes in curated gene sets and gene ontology (GO) terms.

Graph 95.	N Genes	Beta	Beta STD	95% CI for Beta	*p* Value	*p* Value, Bonferroni Correction
GO: CC—GID complex	4	2.48	0.0512	(1.75–3.21)	1.82 × 10^−11^	3.01 × 10^−7^
Reactome—smooth muscle contraction	24	0.827	0.0417	(0.56–1.09)	5.30 × 10^−10^	8.77 × 10^−6^
GO: BP—hard palate development	4	1.8	0.0372	(1.15–2.45)	3.10 × 10^−8^	5.10 × 10^−4^
Reactome—SEMA4D induced cell migration and growth cone collapse	12	0.9	0.0321	(0.57–1.23)	6.53 × 10^−8^	1.08 × 10^−3^
GO: MF—Ferric iron binding	2	2.54	0.0371	(1.58–3.5)	1.07 × 10^−7^	1.77 × 10^−3^
Roylance—Breast cancer 16q copy number down	14	0.969	0.0374	(0.6–1.34)	1.58 × 10^−7^	2.62 × 10^−3^
Reactome—Rho GTPases activate CIT	8	1.01	0.0294	(0.61–1.41)	2.99 × 10^−7^	4.95 × 10^−3^
Reactome—SEMA4D in semaphorin signaling	15	0.72	0.0287	(0.43–1.01)	6.75 × 10^−7^	0.0111
GO: CC—Chromosome telomeric repeat region	7	0.971	0.0265	(0.58–1.37)	7.45 × 10^−7^	0.0123
GO: BP—Negative regulation of synaptic vesicle exocytosis	2	1.98	0.0288	(1.17–2.79)	9.42 × 10^−7^	0.0156
Yih response to arsenite c3	10	0.863	0.0283	(0.51–1.22)	9.62 × 10^−7^	0.0159
GO: BP—Mitotic G1/S transition checkpoint signaling	11	0.98	0.0335	(0.58–1.38)	9.81 × 10^−7^	0.0162
Reactome—Rho GTPases activate ROKs	9	0.979	0.0303	(0.56–1.39)	1.90 × 10^−6^	0.0315
GO: BP—Bone trabecula morphogenesis	8	1.13	0.0328	(0.65–1.61)	2.49 × 10^−6^	0.0411
GO: BP—Neuron neuron synaptic transmission	5	1.48	0.034	(0.85–2.11)	2.50 × 10^−6^	0.0412

Beta STD, standardized regression coefficient; 95% CI, 95% confidence intervals; GO, gene ontology; CC, cellular compartment; BP, biological process; MF, molecular function. GID, glucose-induced degradation deficient; CIT, Citron Rho-Interacting Kinase; SEMA4D, Semaphorin-4D.

**Table 5 nutrients-17-02983-t005:** Adjusted odds ratios and 95% confidence intervals for diabetic retinopathy risk by polygenetic risk scores of the best model (PRS-3SNP) after covariate adjustments according to lifestyles.

	Low-PRS (n = 3880)	Medium-PRS (n = 1038)	High-PRS (n = 192)	*p*-Value for the Interaction of PRS and Lifestyles ^1^
Low fruit High fruit	1 1	9.823 (4.146–23.27) 8.693 (5.200–14.53)	81.32 (32.09–206.1) 47.03 (25.80–85.73)	0.0023
Low fast foods High fast foods	1 1	17.65 (2.180–143.0) 9.021 (5.311 15.32)	95.27 (4.280–193.4) 55.65 (30.80–100.6)	<0.0001
Low coffee High coffee	1 1	8.580 (4.844–15.20) 8.606 (4.270–17.33)	55.38 (29.50–104.0) 34.24 (14.63–80.14)	<0.0001
Low alcohol High alcohol	1 1	8.176 (4.933–13.55) 7.356 (3.106–17.42)	45.71 (26.19–79.79) 40.43 (12.77–128.0)	<0.0001
Low eating duration High eating duration	1 1	10.27 (5.090–20.70) 7.484 (4.328–12.94)	35.27 (15.44–80.60) 56.14 (30.65–102.8)	<0.0001
Low exercise High exercise	1 1	6.893 (1.967 24.15) 9.262 (5.757–14.90)	32.03 (8.275–124.0) 55.77 (31.68–98.18)	<0.0001

The 3-SNPs consisted of adenosine triphosphate (ATP)-binding cassette subfamily A member 4 (*ABCA4*)_rs17110929, matrix metalloproteinase-2 (*MMP2-AS1*)_rs2576531, and forkhead box protein P1 (*FOXP1*)_rs557869288, with its PRS calculated by summing risk alleles of 3 SNPs. Covariates for adjusted logistic regression: age, gender, diabetic duration, BMI, hypertension, serum triglyceride, estimated glomerular filtration rate, smoking, alcohol intake, hypertension, estimated glomerular filtration rate, serum triglyceride, physical activity, lipid-lowering, antihypertensive, and antidiabetic medications. ^1^
*p*-values of the interaction terms (PRS × lifestyle parameter) from a two-way analysis of covariance for DM-DR status adjusted for covariates. References: Fruit: <1 serving/day; fast foods: <1 serving/day; coffee: <0.5 cup/day; alcohol: <3 g/day; eating duration: 15 min/day; exercise: 30 min/day, moderate intensity exercise.

## Data Availability

Data is contained within the article or Appendix A.

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
