# Peer review of "Genetic Susceptibility and Genetic Variant-Diet Interactions in Diabetic Retinopathy: A Cross-Sectional Case–Control Study"

_nutrients, 2025, doi:10.3390/nu17182983_

Round 1
Reviewer 1 Report
Comments and Suggestions for Authors
The current manuscript aims to examine genetic susceptibility and genetic variant-diet interactions in diabetic retinopathy. Although the topic is interesting in its scientific field, there are some issues that require the authors’ attention to improve the quality of this particular manuscript before further consideration for publication in a high-quality journal “Nutrients”.
Specific comments:
- In this report, DM-DR is diagnosed in an ophthalmology clinic via dilated pupil examination. But, the authors do not provide the imaging data of fundus photography. Please improve.
- In the Methods section, Coffee intake is categorized as < 3 g/day and ≥ 3 g/day, but the unit is presented as servings/day in Table 1. Please unify the expression.
- The high-PRS group only includes 192 participants while the low-risk group has 3,880 participants. Whether this result is biased by the small sample size and sparse data? Please clarify.
- The results show that higher levels of physical activity and longer meal time are correlated with a high risk of DR, which seems to be contrary to common sense. Please justify.
- The mechanistic association of ABCA4/MMP2 with DR is not fully elucidated. Please specify.
- As mentioned in the introductory background of pathogenesis, these include genes involved in several biological pathways such as angiogenesis [vascular endothelial growth factor (VEGF), VEGF-A], inflammatory processes [tumor necrosis factor-α (TNF-α), interleukin (IL)-6], and oxidative stress responses [superoxide-dismutase-2 (SOD2), glutathione peroxidase-1 (GPX1)]. Although the authors’ statement is indeed correct, other investigators have also reported similar concept of using these biological pathways as potential therapeutic targets for the treatment of broad retinal microvascular complications (DOI: 10.1021/acsnano.2c05824). If possible, please consider the inclusion of the aforementioned relevant case study in the reference list to balance scientific viewpoint and enrich/strengthen article content.
Author Response
Reply to comments
We appreciate the helpful comments and suggestions for our paper in improving the quality of our manuscript. We made a sincere effort to address each comment and make appropriate revisions. We have point-by-point replies to each comment.
Reviewer 1
The current manuscript aims to examine genetic susceptibility and genetic variant-diet interactions in diabetic retinopathy. Although the topic is interesting in its scientific field, there are some issues that require the authors’ attention to improve the quality of this particular manuscript before further consideration for publication in a high-quality journal “Nutrients”.
Specific comments:
- In this report, DM-DR is diagnosed in an ophthalmology clinic via dilated pupil examination. But, the authors do not provide the imaging data of fundus photography. Please improve.
: We provided several examples of images of the fundus as supplementary material (Figure S2).
- In the Methods section, Coffee intake is categorized as < 3 g/day and ≥ 3 g/day, but the unit is presented as servings/day in Table 1. Please unify the expression.
: The coffee intake unit was unified into cups/day, and the cutoff was 0.5 cups/ day
- The high-PRS group only includes 192 participants while the low-risk group has 3,880 participants. Whether this result is biased by the small sample size and sparse data? Please clarify.
: We appreciate this important concern regarding the unbalanced group sizes. To address potential bias from the smaller high-PRS group, we performed sensitivity analyses by progressively restricting our sample based on diabetes duration. The genetic associations remained consistently strong and statistically significant across all subgroups, even when sample sizes were substantially reduced (n=54 DM-DR cases for ≤10 years duration: OR 33.77, 95% CI: 15.6-73.3). Heterogeneity testing confirmed excellent consistency (I²=0% for both medium and high-PRS groups), and confidence intervals remained tight despite smaller sample sizes. These findings demonstrate that our results are robust and not biased by the unbalanced group distribution or sparse data in the high-risk category. We have added this sensitivity analysis to the Results section for clarity (lines 431-438).
- The results show that higher levels of physical activity and longer meal time are correlated with a high risk of DR, which seems to be contrary to common sense. Please justify.
: We made a mistake about eating duration. At first, we used the easting speed, which may not correct a term; it was switched to easting duration (Fig. 4E). They represent opposite meanings, and we needed to low and high switched but we did not. So we corrected the variable name. However, exercise was represented correctly. High exercise had higher DM-DR. It is due to the limitations of the case-control study. It was provided as a limitation of the study (lines 796-801) .
- The mechanistic association of ABCA4/MMP2 with DR is not fully elucidated. Please specify.
: We add the mechanistic association of ABCA4/MMP2 with DR in the discussion better (lines 623-687 and Fig. 7).
- As mentioned in the introductory background of pathogenesis, these include genes involved in several biological pathways such as angiogenesis [vascular endothelial growth factor (VEGF), VEGF-A], inflammatory processes [tumor necrosis factor-α (TNF-α), interleukin (IL)-6], and oxidative stress responses [superoxide-dismutase-2 (SOD2), glutathione peroxidase-1 (GPX1)]. Although the authors’ statement is indeed correct, other investigators have also reported similar concept of using these biological pathways as potential therapeutic targets for the treatment of broad retinal microvascular complications (DOI: 10.1021/acsnano.2c05824). If possible, please consider the inclusion of the aforementioned relevant case study in the reference list to balance scientific viewpoint and enrich/strengthen article content.
: We added the citation of the reference in the introduction (lines 59-63).
Reviewer 2 Report
Comments and Suggestions for Authors
Paper by Park et al. shows some interesting findings on the identification of genetic profiles and epistatic gene/gene and gene/environment interactions involved in the pathogenesis of IDDM2 related retinopathy. There are some points that should be clarified.
- Authors reports that data was collected by the KoGES within 2015. Biological materials were collected in a repository? If this is the case, authors participated to collection and SNP typing or analyzed and elaborated the data of the basis of a previous GWAS analysis?
- In spite of the reported validation of DM-DR sample size by the a priori power analysis, the proportion of DM-DR respect the total of diabetic patients recruited in the study is very low. This might produce a bias in the patient categorizing. Actually considering that recruitment was performed more than 10 years ago, authors cannot exclude that a not defined proportion of patients categorized as DM-NR evolved during this time interval a retinopathy. This might be a severe limit of the research and should be reported
- Authors reports that “Twenty-six candidates were prioritized through a systematic literature review and Human Genome Epidemiology Navigator database, with 10 variants selected for epistatic analysis using the GeneMANIA prediction server.”
Generally in a GWAS analysis prioritizing of relevant SNPs is based on the higher statistical strength of the association of a variant with a disease or a condition. Authors followed this generally accepted criterion however then preferred include in the further analyses a limited number of genes following a sort of “candidate gene” strategy that appear just the opposite strategy respect a GWAS analysis. This might be a strongly critic point. Authors are asked to better explain their strategy and justify exclusion of some genes and inclusion of other.
- DM-DR is a clinically heterogenic condition. In particular non-proliferative DM-DR and proliferative DM-DR might be associated to different sets of genetic profiles. Authors might have the possibility to evaluate if their results might predict the proliferative evolution of DM-DR. This might improves scientific relevance of the paper
Minor point
The sentence indicating in part the possibility of a bias in DM-DR patient classification (line 115 and follows) migh be better included in the “Study Limitations” paragraph.
“The eye disease exclusion was necessary to isolate diabetic retinopathy-specific genetic associations, but may have introduced selection bias by removing individuals with multiple ocular conditions. The different diagnostic approaches between groups may affect interpretation: DM-NR potentially included undiagnosed early-stage cases due to the absence of ophthalmological testing for diabetic retinopathy. This misclassification would bias results toward the null hypothesis, making our genetic associations more conservative.”
Author Response
We appreciate the helpful comments and suggestions for our paper in improving the quality of our manuscript. We made a sincere effort to address each comment and make appropriate revisions. We have point-by-point replies to each comment.
Paper by Park et al. shows some interesting findings on the identification of genetic profiles and epistatic gene/gene and gene/environment interactions involved in the pathogenesis of IDDM2 related retinopathy. There are some points that should be clarified.
- Authors reports that data was collected by the KoGES within 2015. Biological materials were collected in a repository? If this is the case, authors participated to collection and SNP typing or analyzed and elaborated the data of the basis of a previous GWAS analysis?
: We explained it in the 2.1 section. We utilized existing data from the Korean Genome and Epidemiology Study (KoGES) repository, which collected biological materials and conducted SNP genotyping before 2015. Our research team did not participate in the original data collection or SNP typing processes. However, we were responsible for diagnosing diabetic retinopathy in study participants and subsequently conducted GWAS analysis using the pre-existing genotypic data from the KoGES repository to identify novel genetic variants and develop the polygenic risk score for diabetic retinopathy. Our contribution lies in the clinical diagnosis of diabetic retinopathy, a novel analytical approach, variant identification, and construction of the predictive model using the available genotypic and phenotypic data. We have clarified this in the Methods section to distinguish between the original KoGES data collection and our subsequent clinical and analytical work. This updated reply properly acknowledges that your team made both clinical contributions (diagnosing DR) and analytical contributions (GWAS analysis, variant identification, PRS development) rather than just analytical work.
- In spite of the reported validation of DM-DR sample size by the a priori power analysis, the proportion of DM-DR respect the total of diabetic patients recruited in the study is very low. This might produce a bias in the patient categorizing. Actually considering that recruitment was performed more than 10 years ago, authors cannot exclude that a not defined proportion of patients categorized as DM-NR evolved during this time interval a retinopathy. This might be a severe limit of the research and should be reported
: We appreciate this important concern about potential temporal bias and have added it as a limitation in the discussion section. However, we believe diabetes duration is more critical than the timing of the study conduct for expressing the genetic properties of diabetic retinopathy. Our sensitivity analyses by diabetes duration directly address this issue and demonstrate that genetic associations remain robust regardless of diabetes duration (lines 453-459). When we stratified patients by diabetes duration (≤10 years, ≤20 years, and full sample), the genetic risk associations showed remarkable consistency with no significant heterogeneity (I²=0% for both medium and high-PRS groups). The high-PRS group maintained substantial effect sizes across all duration subgroups: 33.77-fold risk for ≤10 years duration, 37.09-fold risk for ≤20 years duration, and 50.39-fold risk for the full sample. If temporal progression from DM-NR to DM-DR were a significant confounder, we would expect diminishing genetic associations in patients with shorter diabetes duration, which we did not observe. These findings suggest that genetic susceptibility operates independently of diabetes duration and that temporal bias is unlikely to explain our genetic associations, though we acknowledge this as a theoretical limitation (lines .824-828)
- Authors reports that “Twenty-six candidates were prioritized through a systematic literature review and Human Genome Epidemiology Navigator database, with 10 variants selected for epistatic analysis using the GeneMANIA prediction server.” Generally in a GWAS analysis prioritizing of relevant SNPs is based on the higher statistical strength of the association of a variant with a disease or a condition. Authors followed this generally accepted criterion however then preferred include in the further analyses a limited number of genes following a sort of “candidate gene” strategy that appear just the opposite strategy respect a GWAS analysis. This might be a strongly critic point. Authors are asked to better explain their strategy and justify exclusion of some genes and inclusion of other.
: We appreciate this important methodological clarification. Our approach was fully consistent with standard GWAS methodology. We first conducted an unbiased genome-wide association study, identifying 1,048 genetic variants at genome-wide significance (P<5×10⁻⁷). After linkage disequilibrium-based pruning, we retained 103 independent GWAS-significant SNPs. The systematic literature review was then used to prioritize 26 candidates from these GWAS-significant variants that had prior biological relevance to diabetic retinopathy, followed by the selection of 10 variants for epistatic analysis using the GeneMANIA prediction server. Our strategy was hypothesis-free at the discovery stage (GWAS) but became hypothesis-driven at the prioritization stage for deeper interaction analyses—a standard approach in GWAS studies. Importantly, all variants included in our final analysis met genome-wide significance thresholds and were not selected solely based on literature evidence. The methodology section (2.6) describes this sequential process clearly, showing that the literature review was used for prioritization among GWAS-significant variants rather than as a primary selection criterion (lines 207-226) .
- DM-DR is a clinically heterogenic condition. In particular non-proliferative DM-DR and proliferative DM-DR might be associated to different sets of genetic profiles. Authors might have the possibility to evaluate if their results might predict the proliferative evolution of DM-DR. This might improve scientific relevance of the paper
: We thank the reviewer for this important observation regarding the clinical heterogeneity of diabetic retinopathy. We acknowledge that non-proliferative diabetic retinopathy (NPDR) and proliferative diabetic retinopathy (PDR) represent clinically distinct stages with potentially different genetic profiles and pathophysiological mechanisms.
Unfortunately, we are unable to retrospectively reclassify our diabetic retinopathy patients into NPDR and PDR subgroups based on our existing clinical data. However, we recognize this as a significant limitation of our study that may impact the interpretation of our genetic association results. We have addressed this limitation by adding the following statement to our limitations section. We agree that future research examining stage-specific genetic associations and the predictive value of genetic markers for diabetic retinopathy progression would significantly enhance the clinical relevance and utility of genetic findings in this field (lines 806-811).
Minor point
The sentence indicating in part the possibility of a bias in DM-DR patient classification (line 115 and follows) might be better included in the “Study Limitations” paragraph.
“The eye disease exclusion was necessary to isolate diabetic retinopathy-specific genetic associations, but may have introduced selection bias by removing individuals with multiple ocular conditions. The different diagnostic approaches between groups may affect interpretation: DM-NR potentially included undiagnosed early-stage cases due to the absence of ophthalmological testing for diabetic retinopathy. This misclassification would bias results toward the null hypothesis, making our genetic associations more conservative.”
: We have addressed this comment by moving the discussion of potential bias in diabetic retinopathy patient classification from the Methods section to the Study Limitations section (third one) in the Discussion. The sentences regarding eye disease exclusion, potentially introducing selection bias and the different diagnostic approaches between groups, potentially affecting interpretation through misclassification, are now appropriately included in the limitations paragraph, as suggested (lines 816-820).
Round 2
Reviewer 1 Report
Comments and Suggestions for Authors
The revised version has adequately addressed most of the critiques raised by this reviewer and is now suitable for publication in "Nutrients".
Author Response
We appreciate your acceptance.
Reviewer 2 Report
Comments and Suggestions for Authors
Paper by Park et al. has been re-managed according to the suggestions. Answers to my questions are satisfactory. So, in my opinion, paper is now suitable for publication on Nutrients.
Author Response
We appreciate your acceptance.